# Patient education for older adults with cancer and their caregivers: Protocol for a scoping review

Lu Lin[1,2], Janet Papadakos[3], Rouhi Fazelzad[4], Kristen Haase[5], Shabbir Alibhai[6], Fay Bennie[7], Martine Puts[8]*

1 The First Affiliated Hospital of Soochow University, Suzhou, Jiangsu, China, 2 Lawrence Bloomberg Faculty of Nursing, University of Toronto, Toronto, Ontario, Canada, 3 The Centre for Patient Education, Princess Margaret Cancer Centre, and Dalla Lana School of Public Health, University of Toronto, Toronto, Ontario, Canada, 4 Library and Information Services, Princess Margaret Cancer Centre, University Health Network, Toronto, Ontario, Canada, 5 School of Nursing, University of British Columbia, Vancouver, British Columbia, Canada, 6 Department of Supportive Care, Princess Margaret Cancer Centre, University Health Network, and Faculty of Medicine and Institute for Health Policy, Management and Evaluation, University of Toronto, Toronto, Ontario, Canada, 7 Older adult team member, Dunsford, Ontario, Canada, 8 Lawrence Bloomberg Faculty of Nursing, University of Toronto and Department of Supportive Care, Princess Margaret Cancer Centre, University Health Network, Toronto, Ontario, Canada

* martine.puts@utoronto.ca

## Abstract

Treatment decision-making for older adults is complex due to multimorbidity, polypharmacy, and age-related changes such as sensory impairment and cognitive decline. In addition, many older adults have lower health literacy, which impacts their ability to access, understand, and apply health information. Patient education plays a critical role to comprehend a new cancer diagnosis, explore treatment options, and manage care effectively. To be effective, education materials must be specifically tailored to the needs of this population. Despite growing recognition of the importance of geriatric-specific cancer education, no recent review has synthesized the literature on educational interventions or materials developed for this population. This review aims to answer the question: What is known about patient education interventions and materials developed for older adults with cancer and their caregivers and their effectiveness to improve outcomes?. We will follow the Arksey and O'Malley framework, refined by Levac et al., and report according to the PRISMA-ScR checklist. We will search Medline ALL, Embase, Emcare, Cochrane Central, PsycInfo (all via OvidSP), and Scopus (via Elsevier) for studies published from 2000 onward and complete a manual search of the references of included studies. A health sciences librarian conducted the searches. We will include peer-reviewed and relevant grey literature. Stakeholder consultation will be conducted to inform interpretation of findings and identify gaps. Eligible studies will use qualitative or quantitative designs focused on geriatric oncology patient education. Two reviewers will independently screen studies and extract data using Covidence and Excel. Data will be analyzed using

**Data availability statement:** No datasets were generated or analysed during the current study. All relevant data from this study will be made available upon study completion.

**Funding:** The author(s) received no specific funding for this work.

**Competing interests:** The authors have declared that no competing interests exist.

descriptive statistics and narrative synthesis. The quality of peer-reviewed studies will be appraised with the Mixed Methods Appraisal Tool (2018) for descriptive purposes only. This scoping review will map existing educational interventions and materials for older adults with cancer, support clinicians in practice, and highlight gaps to inform future research and development.

## Introduction

Cancer is a disease that mostly affects older adults [1]. In most countries around the world the number of older adults diagnosed with cancer will increase significantly in the next years [1]. Older adults often have comorbid conditions, polypharmacy, falls and other geriatric syndromes, that could impact cancer treatment risks and benefits [2]. However, these older adults are underrepresented in clinical research and thus there is little evidence to guide treatment decision making by the clinical team [3]. The American Society of Clinical Oncology has published geriatric oncology guidelines recommending that every older adult aged 65 years and over considered for any cancer treatment should have a geriatric assessment to guide the cancer treatment and supportive care plan [2]. Several surveys have shown knowledge gaps among the various oncology providers in terms of how to conduct a comprehensive geriatric assessment [4–8].

To reduce this knowledge gap, many geriatric oncology educational modules target providers showing them how to conduct geriatric assessments [9,10]. A previous systematic review on therapeutic education for older adults with cancer with a search conducted of papers published between 1990 and 2016 showed only 14 papers and of those only 7 targeted older adults [11]. In this review, education was part of a larger intervention that included symptom management, exercises or rehabilitation or medication adherence intervention, but none seemed to address specific geriatric oncology knowledge for older adults with cancer and the education was not used as a standalone intervention, it was combined with follow-up by health care provider support. However, this review is older, and an updated review is needed [11].

Patient education is important to enhance health literacy. The WHO defines health literacy as "representing the personal knowledge and competencies that accumulate through daily activities, social interactions and across generations. Personal knowledge and competencies are mediated by the organizational structures and availability of resources that enable people to access, understand, appraise, and use information and services in ways that promote and maintain good health and well-being for themselves and those around them." Health literacy plays a crucial role in enhancing individuals' ability to access, comprehend, and effectively utilize reliable health information. It empowers people to make informed personal health decisions and actively participate in collective health promotion efforts to address key health determinants [12]. Older adults may have lower levels of health literacy and ehealth literacy that can impact shared decision making as well as self-management during the treatment [13]. Low health literacy is associated with poorer health outcomes

and difficulties following medication instructions and less able to make informed health decisions [14]. Patient education is essential for helping older adults and their caregivers understand a new cancer diagnosis, treatment options, and the self-management required to enhance quality of life [11,15–19]. However, due to age-related factors (such as sensory and cognitive changes), the educational needs of older cancer patients and their caregivers can differ significantly from those of younger patients [20–25]. By reviewing current evidence on educational interventions designed for older adults newly diagnosed with cancer and their caregivers, this scoping review aims to map the existing research landscape and identify effective educational content, delivery methods and the intended outcomes, to better support this population and inform future research. Therefore, our scoping review question is: What is known about patient education interventions and materials developed for older adults with cancer and their caregivers and their effectiveness to improve outcomes? We aim to map and synthesize evidence on educational interventions and materials designed specifically for older adults with cancer and their caregivers, including the type of educational content, delivery formats, and effectiveness of the education on outcomes. We will also identify gaps in current knowledge to inform future research and educational development.

## Methods

This scoping review will be conducted following the framework proposed by Arksey and O'Malley [26], with subsequent refinements by Levac et al.[27]. The reporting will adhere to the Preferred Reporting Items for Systematic Reviews and Meta-Analyses extension for Scoping Reviews (PRISMA-ScR) [28]. Additionally, the PRISMA-Protocol (PRISMA-P) statement was used to guide the development of this protocol [29]. The completed PRISMA-P Checklist is provided in S1 File.

### Inclusion and exclusion criteria

Eligibility criteria are structured based on the PICOS (Population, Intervention, Comparison, Outcome, Study Design) framework.

**Population** Studies examining patient education materials for older adults (aged 65 and above) with cancer and their care partners in any clinical oncology setting (inpatient or outpatient) will be included. Studies will also be eligible if the mean participant age is over 65, if subgroup analyses for older adults are reported, or if the population is explicitly described as an older adult group.

**Intervention** Patient education materials (modules/websites/written materials/audiovisuals) targeting older adults with cancer and their caregivers and taking into account physical, psychological and social changes with aging and how that impacts cancer treatment risk and benefits, outcomes and treatment delivery. Types of Patient Educational interventions can include [14–18]:

- Therapeutic patient education (defined as education aimed at enabling the patient and their relatives to cope with the disease in partnership with health professionals [11])

- Psychosocial or behavioral education

- Counseling with an educational component related to cancer management

- Education delivered through various formats, including individual sessions, group classes, or digital/telehealth platforms.

**Comparison**: it can be any/no comparator.
**Outcomes:** Any relevant outcome [6,19,20], such as:

- User data on the tool

- Knowledge improvement and retention

- Psychological outcomes (e.g., anxiety, depression, empowerment)

- Health-related quality of life

- Shared decision-making, decisional regrets

- Treatment adherence

- Caregiver satisfaction, burden, or stress.

**Study design**: All study designs will be considered eligible. Additionally, studies focusing solely on caregivers of older adults with cancer will also be included.

### Exclusion criteria

- Conference abstracts, theses, reviews, editorials, and commentaries will be excluded. However, reviews will be examined to identify any original studies that meet the inclusion criteria. For conference abstracts, study authors will be contacted to confirm whether a full study has been published.

- Studies published before 2000 will be excluded. Recognition of the unique needs of older adults with cancer began to emerge in the 1990s, and the past two decades have seen significant advancements in cancer treatment tailored to this population. As a result, patient education materials developed prior to 2000 are likely to be less relevant to current clinical practice.

   Studies that evaluate combined interventions, where the effects of patient education cannot be isolated from other components (e.g., psychological counseling), will be excluded.

### Search strategy

In collaboration with an information specialist, a comprehensive literature search was conducted from 2000 to December 2024, in Medline ALL (Medline and Medline Epub Ahead of print and In-Process & Other Non-Indexed Citations), Embase, Emcare, Cochrane Central Register of Controlled Trials, and PsycInfo all from the OvidSP platform, and Scopus from Elsevier. Controlled vocabulary terms and text words adapting the database-specific search syntax will be used where available. Where applicable, the search will be restricted to human studies, and older adults based on each database-indexed age group, and exclude books, and preprints. There will be no language restrictions. The reference lists from included studies, scoping and systematic review studies will be screened to identify additional potentially relevant studies. Please see S2 File for the preliminary Medline search.

### Grey literature

To identify additional relevant geriatric oncology patient education studies not captured through database searches, we will conduct a grey literature search. This will include reviewing websites of patient support organizations, the American Society of Clinical Oncology, the European Society of Medical Oncology, and various oncology nursing societies. In addition, we will include dissertations that meet the eligibility criteria, as they may contain original and relevant data not yet published in peer-reviewed journals.

### Study selection and screening

Following deduplication in EndNote, references will be imported into the Covidence software [30] for further screening. Covidence's deduplication function will also be used. Two reviewers will independently screen titles and abstracts for eligibility using Covidence. Full texts of selected studies will then be assessed by two independent reviewers based on the eligibility criteria. Any discrepancies during abstract and full-text screening will be resolved by a third reviewer. To ensure

consistency, each reviewer will initially screen 50 abstracts, followed by a team discussion to address any conflicts or ambiguities before proceeding with the remaining titles and abstracts.

## Data extraction

Data from eligible studies will be extracted using a standardized Microsoft Excel spreadsheet, capturing participant demographics, details of patient education materials (e.g., aim, form, content etc., theoretical framework used), control conditions, outcomes, and study design. The data extraction form, developed by the research team, is available in S3 File. The form will be piloted by two reviewers using the first five studies, with adjustments made if necessary. If any information is missing from the publication, study authors will be contacted via email, with a follow-up reminder sent after two weeks if no response is received.

## Quality assessment

The quality of included peer-reviewed studies will be assessed using the 2018 version of the Mixed Methods Appraisal Tool (MMAT) [31,32] within an Excel spreadsheet, conducted alongside data extraction to provide an overview of methodological quality for descriptive purposes only. Grey literature will be included for mapping but not formally appraised. Two independent reviewers will perform the assessment, with any discrepancies resolved by a third reviewer. Studies will not be excluded based on quality assessment; rather, this process will provide an overview of the quality of the available evidence.

## Summarizing and reporting results

Results will be reported following PRISMA-ScR [28], and a PRISMA flow diagram will illustrate the review process. The quality assessment findings will be summarized in a table. Data will be analyzed and reported through a combination of descriptive numerical summaries and narrative analysis. A descriptive summary will highlight the different geriatric oncology education interventions and materials developed and tested, along with the associated outcomes, including the type of educational content, delivery formats, and effectiveness of the education on outcomes. Additionally, an outcome mapping table will categorize the primary outcome types related to patient education interventions across the included studies.

## Consultation

A consultation will be conducted with patients, caregivers, and clinicians involved in geriatric oncology, consistent with the optional sixth step of the scoping review methodology as proposed by Arksey and O'Malley [26] and further developed by Levac et al. [27]. This step is intended to enhance the relevance and applicability of the review findings, rather than to contribute to data collection or the narrative synthesis. Participants will be presented with a summary of the preliminary results of the review and invited to provide feedback on the comprehensiveness, clarity, and practical implications of the findings. Their input will help validate the results, highlight knowledge gaps not captured in the literature, and will inform recommendations for future research and development of patient education resources. The consultation will be conducted via Zoom and is not part of the formal data analysis. The feedback will be summarized anonymously.

## Ethics statement

No personal or identifying data will be collected during this scoping review. As such, research ethics board (REB) approval is not required.

## Timeline

The literature search was conducted on January 27 2025. The first draft is expected to be completed by September 2025, with the final version prepared for journal submission by December 2025.

## Discussion

This scoping review is intended to summarize the state of patient education interventions and materials developed for older adults with cancer and their caregivers, including educational content, delivery methods and outcomes, and to identify knowledge gaps. These findings will be useful for informing future educational interventions and development of age-appropriate education materials for this population. The number of older adults is increasing everywhere around the world, and due to age-related changes, the older adult population has unique information needs. In addition, treatment risks and benefits may be different for older adults with cancer compared to younger patient with cancer due to multimorbidity, polypharmacy, age-related decline in physiological reserve which all impact the treatment risks and in the context of declining life expectancy, benefits from treatments. In addition, many older adults with cancer rely on a support person to help them with cancer-treatment decision-making and during the treatments.

The scoping review will explore and summarize information from previously published literature. It will not involve patients or the collection of primary research, therefore ethics approval is not required. Normally as part of the Arksey and O'Malley scoping review framework, a consultation with stakeholders is recommended [26] and this is planned for this review.

Dissemination of the scoping review results to the broad scientific community will take place through the publication of a manuscript and presentation of results at scientific and clinical conferences.

A strength of our scoping review is a comprehensive search approach including multiple databases, multiple languages and inclusion of all types of studies. A limitation is that some of the interventions that we will identify may be older and less relevant to the current clinical setting but f so we can learn from the previous studies what the next steps for geriatric oncology education should be.

## Conclusion

The results of the scoping review will provide a summary of the literature regarding geriatric oncology patient education interventions which can be used by healthcare providers with older adults with cancer.

## Supporting information

**S1 File. PRISMA-P checklist.docx**
(DOCX)

**S2 File. Preliminary Medline search.**
(DOCX)

**S3 File. Data extraction form.**
(DOCX)

## Author contributions

**Conceptualization:** Lu Lin, Janet Papadakos, Rouhi Fazelzad, Kristen Haase, Shabbir Alibhai, Fay Bennie, Martine Puts.

**Methodology:** Lu Lin, Janet Papadakos, Rouhi Fazelzad, Kristen Haase, Shabbir Alibhai, Fay Bennie, Martine Puts.

**Project administration:** Lu Lin, Martine Puts.

**Supervision:** Martine Puts.

**Writing – original draft:** Lu Lin, Martine Puts.

**Writing – review & editing:** Lu Lin, Janet Papadakos, Rouhi Fazelzad, Kristen Haase, Shabbir Alibhai, Fay Bennie.

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
