## [Decision Letter · Decision Letter 0]

PONE-D-25-04797Patient Education for Older Adults with Cancer and Their Caregivers: Protocol for a Scoping ReviewPLOS ONE

Dear Dr. Puts,

Thank you for submitting your manuscript to PLOS ONE. After careful consideration, we feel that it has merit but does not fully meet PLOS ONE’s publication criteria as it currently stands. Therefore, we invite you to submit a revised version of the manuscript that addresses the points raised during the review process.

I decided to make a decision after one review as the reviewer has highlighted the main aspects for revision of the principally suitable review protocol. Please revise the manuscript considering the reviewer’s suggestions.

In addition to the reviewer's comments some clarifications are needed: (1) I do not see why a “quality assessment” is planned as this is not recommended for scoping reviews. Also, as pointed out by the reviewer, this might not be the most suitable tool. (2) The reviewer (although erroneously referring to 5 years) is correct in questioning the reason for limiting the search to papers published since 2000. Although, I think this limitation is feasible, the reason should rather be the quality of the materials than the foundation of the society. (3) As stated by the reviewer, the abstract fails to mention the search for grey literature and the planned consultation. I find it surprising that e.g. dissertations are excluded while information from websites will be included. Please explain. The purpose of the consultation is not clear. Is this part of the search for “grey literature” or of the narrative summary or will the participants be presented the results of the review and asked to draw conclusions and give recommendations from the results? This should either be deleted or reported in more detail. (4) Also the “narrative summary” should be reported in more detail. What do you anticipate that will be reported, e.g. types  of materials, content of materials etc.? Are you planning to use any specific methods (cf. the Cochrane handbook chapter on narrative syntheses). (5) Although Prisma-ScR is given as reporting guideline, the appendix presents Prisma-P. Please clarify. (6) In line 103, please exchange pe by patient education.

We look forward to receiving your revised manuscript.

Kind regards,

Sascha Köpke

Academic Editor

PLOS ONE

Additional Editor Comments (if provided):

Reviewers' comments:

Reviewer's Responses to Questions

**Comments to the Author**

1. Does the manuscript provide a valid rationale for the proposed study, with clearly identified and justified research questions?

Reviewer #1: Partly

2. Is the protocol technically sound and planned in a manner that will lead to a meaningful outcome and allow testing the stated hypotheses?

Reviewer #1: Partly

3. Is the methodology feasible and described in sufficient detail to allow the work to be replicable?

Reviewer #1: No

4. Have the authors described where all data underlying the findings will be made available when the study is complete?

Reviewer #1: Yes

5. Is the manuscript presented in an intelligible fashion and written in standard English?

Reviewer #1: Yes

6. Review Comments to the Author

You may also provide optional suggestions and comments to authors that they might find helpful in planning their study.

Reviewer #1: Dear authors,

thank you for introducing this interesting manuscript on “Patient Education for Older Adults with Cancer and Their Caregivers: Protocol for a Scoping Review”. The topic is contemporary and important, however the manuscript still requires revisions.

Here are my recommendations for revisions

The purpose of the review needs to be clarified. The introduction chapter is written so that there is some unclarity is the review aiming to fill in gaps of health care providers knowledge or is it providing summary on what constitutes quality patient education tailored to needs of older adults with cancer based on previous literature. Please, review critically the introduction chapter for clarity. In the discussion or conclusion, I would expect you to come back on the knowledge gaps, if this review is aiming also to fill in those (how will the results be used).

Abstract refers to scoping review that combines studies with different methodologies, however, in the manuscript method section, grey literature and consultation of patients, caregivers and clinicians is introduced. Please revise abstract accordingly.

The research questions is broad and varies in the manuscript. What is known about patient educational materials that have been developed for older adults with cancer, aims to identify and synthesize evidence on patient education (educational content, preferred delivery methods, and intended outcomes). This raises question on focus such as is the interest also on effectiveness of interventions and instruments used to assess the effect or impact etc. In the abstract and beginning of discussion, there is mention that focus is on materials, however, later in the manuscript it is also mentioned that review has focus on interventions (is this the delivery methods?). Which is it? Please review purpose and aims to ensure logic and clarity.

The reasons given for 5 years limit is explained, however, I am not sure that establishment year of a society is a reason, and moreover, has there been major development on patient education of the population just in 5 years, as mentioned there has been interest since of 1990’, and patient education from general view (such as empowering patient education) has been research interest for longer time than just 5 years.

Although actual research data is not collected on the Zoom calls with patients, caregivers and clinicians indicating that ethical review process is not needed, there is ethical matters to discuss (such as data privacy etc.).

Quality assessment. If you intent to use the MMAT which is for mixed methods studies. how suitable it is for the grey literature?

I hope you will work on the manuscript further and resubmit it again.

7. PLOS authors have the option to publish the peer review history of their article (what does this mean? ). If published, this will include your full peer review and any attached files.

**Do you want your identity to be public for this peer review?** For information about this choice, including consent withdrawal, please see our Privacy Policy .

Reviewer #1: No

---

## [Author Response · Author response to Decision Letter 1]

12 May 2025

Dear Dr. Köpke,

We would like to thank you and the reviewer for the time spent on the manuscript, we have revised the manuscript according to the suggestions. Below we have provided a point-by-point response to each comment raised.

With kind regards,

Martine Puts on behalf of all authors

Editor’s Comments:

I do not see why a “quality assessment” is planned as this is not recommended for scoping reviews. Also, as pointed out by the reviewer, this might not be the most suitable tool.

We acknowledge that quality assessment is not a required element of scoping reviews but optional. However, we have included it to provide an overview of the methodological rigor of included studies to be able to formulate clearer recommendations for future research based on the quality of the evidence. We have clarified this rationale in the revised protocol in the abstract and in the Methods section under “Quality assessment”. Regarding tool suitability, we have reconsidered the use of the MMAT and now indicate that it will be used solely for descriptive purposes, without influencing inclusion decisions. Eligible grey literature will be included but will not be formally appraised using the MMAT or any other tool.

Revised text abstract on page 3:

The quality of peer-reviewed studies will be appraised with the Mixed Methods Appraisal Tool (2018) for descriptive purposes only.

Revised text in methods section on page 12:

The quality of included peer-reviewed studies will be assessed using the 2018 version of the Mixed Methods Appraisal Tool (MMAT) (31, 32) within an Excel spreadsheet, conducted alongside data extraction to provide an overview of methodological quality for descriptive purposes only.

The reviewer (although erroneously referring to 5 years) is correct in questioning the reason for limiting the search to papers published since 2000. Although, I think this limitation is feasible, the reason should rather be the quality of the materials than the foundation of society.

Thank you for this insightful comment. We have removed the reference to the establishment of the International Society of Geriatric Oncology in 2000 as justification for the publication time restriction. Instead, we now emphasize that the decision to include studies from 2000 onward is based on the increasing attention to geriatric-specific patient education and the improved quality and relevance of materials developed in the past two decades.

Revised text on page 9:

Recognition of the unique needs of older adults with cancer began to emerge in the 1990s, and the past two decades have seen significant advancements in cancer treatment tailored to this population. As a result, patient education materials developed prior to 2000 are likely to be less relevant to current clinical practice.

As stated by the reviewer, the abstract fails to mention the search for grey literature and the planned consultation.

We have revised the abstract to include both the grey literature search and the planned consultation.

New text on page 4:

We will include peer-reviewed and relevant grey literature. Stakeholder consultation will be conducted to inform interpretation of findings and identify gaps.

I find it surprising that e.g. dissertations are excluded while information from websites will be included. Please explain.

Initially, we excluded dissertations as we assumed many would already be published in peer-reviewed journals. However, we acknowledge the value that dissertations can add, particularly in capturing emerging or underreported work. In response to your suggestion, we have revised our inclusion criteria to allow dissertations that meet our eligibility standards.

Revised text on page 4 (Abstract):

We will include peer-reviewed and relevant grey literature. Stakeholder consultation will be conducted to inform interpretation of findings and identify gaps.

Revised text in methods on page 10:

In addition, we will include dissertations that meet the eligibility criteria, as they may contain original and relevant data not yet published in peer-reviewed journals.

The purpose of the consultation is not clear. Is this part of the search for “grey literature” or of the narrative summary or will the participants be presented the results of the review and asked to draw conclusions and give recommendations from the results? This should either be deleted or reported in more detail.

We have revised the manuscript to clarify the purpose of the consultation. The consultation is an optional sixth step of the scoping review methodology, as recommended by Arksey and O’Malley and expanded by Levac et al. Its purpose is to present the preliminary findings of the review to patients, caregivers, and clinicians in geriatric oncology, and to gather their feedback on the relevance, completeness, and practical implications of the results. This feedback will help identify knowledge gaps and inform directions for future research and educational development.

Revised text on page 13:

A consultation will be conducted with patients, caregivers, and clinicians involved in geriatric oncology, consistent with the optional sixth step of the scoping review methodology as proposed by Arksey and O’Malley and further developed by Levac et al. This step is intended to enhance the relevance and applicability of the review findings, rather than to contribute to data collection or the narrative synthesis. Participants will be presented with a summary of the preliminary results of the review and invited to provide feedback on the comprehensiveness, clarity, and practical implications of the findings. Their input will help validate the results, highlight knowledge gaps not captured in the literature, and inform recommendations for future research and development of patient education resources. The consultation will be conducted via Zoom and is not part of formal data analysis. No personal or identifying data will be collected, and participation will be voluntary. As such, research ethics board (REB) approval is not required, but ethical practices, including informed consent, data privacy, and anonymization of feedback, will be followed.

Also the “narrative summary” should be reported in more detail. What do you anticipate that will be reported, e.g. types of materials, content of materials etc.? Are you planning to use any specific methods (cf. the Cochrane handbook chapter on narrative syntheses).

We have expanded the relevant section to clarify that the narrative summary will describe the type of educational content, delivery formats, and effectiveness of the education on outcomes. Although we do not plan to conduct a formal narrative synthesis as outlined in the Cochrane Handbook, we will apply a thematic grouping approach to organize and report the findings, highlighting trends, variations, and gaps across the included studies.

Revised text on page 12:

Data will be analyzed and reported through a combination of descriptive numerical summaries and narrative analysis. A descriptive summary will highlight the different geriatric oncology education interventions and materials developed and tested, along with the associated outcomes, including the type of educational content, delivery formats, and effectiveness of the education on outcomes.

Although Prisma-ScR is given as reporting guideline, the appendix presents Prisma-P. Please clarify.

Thank you for your observation. As this manuscript is only a protocol for a scoping review that has not yet been completed, we included the PRISMA-P checklist in the Appendix to guide protocol development. Once the review is completed, the results will be reported in accordance with the PRISMA-ScR guideline, as stated in the Methods section.

In line 103, please exchange pe by patient education.

We have corrected the phrase in line 103 from “pe” to “patient education”.

Reviewer #1:

Dear authors,

thank you for introducing this interesting manuscript on “Patient Education for Older Adults with Cancer and Their Caregivers: Protocol for a Scoping Review”. The topic is contemporary and important, however the manuscript still requires revisions.

Here are my recommendations for revisions

The purpose of the review needs to be clarified. The introduction chapter is written so that there is some unclarity is the review aiming to fill in gaps of health care providers knowledge or is it providing summary on what constitutes quality patient education tailored to needs of older adults with cancer based on previous literature. Please, review critically the introduction chapter for clarity. In the discussion or conclusion, I would expect you to come back on the knowledge gaps, if this review is aiming also to fill in those (how will the results be used).

We have revised the Introduction and Discussion sections to clarify that the primary aim of this scoping review is to map the characteristics of patient education interventions and materials developed for older adults with cancer and their caregivers, including the type of educational content, delivery formats, and effectiveness of the education on outcomes. While we are not directly evaluating healthcare provider knowledge gaps, we seek to identify gaps in the existing literature to inform future research and development of tailored educational interventions.

Revised text on page 7 (Introduction)

We aim to map and synthesize evidence on educational interventions and materials designed specifically for older adults with cancer and their caregivers, including the type of educational content, delivery formats, and effectiveness of the education on outcomes. We will also identify gaps in current knowledge to inform future research and educational development.

Revised text on page 13-14 (Discussion)

This scoping review is intended to summarize the state of patient education interventions and materials developed for older adults with cancer and their caregivers, including educational content, delivery methods and outcomes, and to identify knowledge gaps. These findings will be useful for informing future educational interventions and development of age-appropriate education materials for this population.

Abstract refers to scoping review that combines studies with different methodologies, however, in the manuscript method section, grey literature and consultation of patients, caregivers and clinicians is introduced. Please revise abstract accordingly.

We have revised the abstract to include both the grey literature search and the planned consultation.

The research questions is broad and varies in the manuscript. What is known about patient educational materials that have been developed for older adults with cancer, aims to identify and synthesize evidence on patient education (educational content, preferred delivery methods, and effectiveness of the education on outcomes). This raises question on focus such as is the interest also on effectiveness of interventions and instruments used to assess the effect or impact etc. In the abstract and beginning of discussion, there is mention that focus is on materials, however, later in the manuscript it is also mentioned that review has focus on interventions (is this the delivery methods?). Which is it? Please review purpose and aims to ensure logic and clarity.

We appreciate this observation. The research question and objectives have now been revised for consistency across the Abstract, Introduction, and Methods. The research question is stated as: What is known about patient education interventions and materials developed for older adults with cancer and their caregivers and their effectiveness to improve outcomes? We aim to identify and synthesize published evidence on the content, delivery methods, and outcomes of patient education interventions and materials designed specifically for older adults with cancer and their caregivers. Where applicable, we will also extract information on the effectiveness of educational interventions, if such data are reported in the studies included.

The reasons given for 5 years limit is explained, however, I am not sure that establishment year of a society is a reason, and moreover, has there been major development on patient education of the population just in 5 years, as mentioned there has been interest since of 1990’, and patient education from general view (such as empowering patient education) has been research interest for longer time than just 5 years.

We have removed the reference to the establishment of the International Society of Geriatric Oncology in 2000 as justification for the publication time restriction. Instead, we now emphasize that the decision to include studies from 2000 onward is based on the increasing attention to geriatric-specific patient education and the improved quality and relevance of materials developed in the past two decades.

Although actual research data is not collected on the Zoom calls with patients, caregivers and clinicians indicating that ethical review process is not needed, there is ethical matters to discuss (such as data privacy etc.).

We agree that, although formal ethics approval is not required, ethical considerations remain important. In response, we have revised the “Ethical Considerations” section to clarify that participation will be entirely voluntary and that ethical practices, including informed consent, data privacy, and anonymization of feedback, will be strictly followed to ensure transparency and respect for participants.

Quality assessment. If you intent to use the MMAT which is for mixed methods studies. how suitable it is for the grey literature?

We agree that the MMAT is not appropriate for grey literature. We have clarified in the Quality Assessment section that only peer-reviewed studies will undergo quality appraisal using MMAT. Eligible grey literature will be included and mapped but will not be formally appraised using the MMAT or any other tool.

I hope you will work on the manuscript further and resubmit it again.

We sincerely appreciate your time and valuable insights. In response, we have critically revised the manuscript to improve clarity, consistency, and methodological rigor.

---

## [Decision Letter · Decision Letter 1]

Patient Education for Older Adults with Cancer and Their Caregivers: Protocol for a Scoping Review

PONE-D-25-04797R1

Dear Dr. Puts,

We’re pleased to inform you that your manuscript has been judged scientifically suitable for publication and will be formally accepted for publication once it meets all outstanding technical requirements.

Kind regards,

Sascha Köpke

Academic Editor

PLOS ONE

Reviewers' comments:

Reviewer's Responses to Questions

**Comments to the Author**

1. Does the manuscript provide a valid rationale for the proposed study, with clearly identified and justified research questions?

Reviewer #1: Yes

2. Is the protocol technically sound and planned in a manner that will lead to a meaningful outcome and allow testing the stated hypotheses?

Reviewer #1: Yes

3. Is the methodology feasible and described in sufficient detail to allow the work to be replicable?

Reviewer #1: Yes

4. Have the authors described where all data underlying the findings will be made available when the study is complete?

Reviewer #1: Yes

5. Is the manuscript presented in an intelligible fashion and written in standard English?

Reviewer #1: Yes

6. Review Comments to the Author

You may also provide optional suggestions and comments to authors that they might find helpful in planning their study.

Reviewer #1: Thank you for resubmitting the manuscript on “Patient Education for Older Adults with Cancer and Their Caregivers: Protocol for a Scoping Review”. I am satisfied with the revisions completed.

7. PLOS authors have the option to publish the peer review history of their article (what does this mean? ). If published, this will include your full peer review and any attached files.

**Do you want your identity to be public for this peer review?** For information about this choice, including consent withdrawal, please see our Privacy Policy .

Reviewer #1: **Yes: ** Virpi Sulosaari

---

## [Editor Report · Acceptance letter]

PONE-D-25-04797R1

PLOS ONE

Dear Dr. Puts,

I'm pleased to inform you that your manuscript has been deemed suitable for publication in PLOS ONE. Congratulations! Your manuscript is now being handed over to our production team.

Kind regards,

on behalf of

Professor Sascha Köpke

Academic Editor

PLOS ONE